# Evaluation of the Occurrence of *Staphylococcaceae* with Reduced Susceptibility to Cefoxitin in Wild Ungulates in Brandenburg, Germany, Based on Land Use-Related Factors

Rafael H. Mateus-Vargas,[a] Tobias Lienen,[b] Denny Maaz,[a] Martin Richter,[b] Sven Maurischat,[b] Julia Steinhoff-Wagner[a,c]

aGerman Federal Institute for Risk Assessment, Department Safety in the Food Chain, Berlin, Germany
bGerman Federal Institute for Risk Assessment, Department Biological Safety, Berlin, Germany
cTechnical University of Munich, School of Life Sciences, Munich, Germany

**ABSTRACT** Interactions between natural and human-used environments have a significant influence on the spread of antimicrobial resistance in wild ecosystems. Despite current knowledge, fundamental questions about the degree of impact of land use-related factors on the spread of antimicrobial-resistant staphylococci in European wild game animal populations have not yet been answered with certainty. In this study, we evaluated the occurrence of *Staphylococcaceae* showing reduced susceptibility to cefoxitin in nasal swabs of fallow deer (*Dama dama*), red deer (*Cervus elaphus*), roe deer (*Capreolus capreolus*), and wild boar (*Sus scrofa*) hunted in Brandenburg, Germany. Evaluations were focused on the use of open-source data regarding the extent as well as the degree of land use, especially for settlement or animal husbandry. Results showed that the detection rate of *Staphylococcaceae* showing a non-wild-type phenotype for cefoxitin differed between animal species of the studied hunting districts. Statistical analyses of results combined with data on land use features revealed that a high density of cattle or poultry in a county may be associated with an increased detection rate in roe deer or wild boar, respectively. Furthermore, positive correlations were determined between the prevalence of non-wild-type *Staphylococcaceae* in roe deer or fallow deer and the proportional extent of surface water bodies in the corresponding area. The presented approach establishes a general basis for a risk-oriented assessment of the effects of human activities on the epidemiology of transmissible microorganisms in the human-animal-environment interface, including antimicrobial-resistant bacteria.

**IMPORTANCE** Intensive research regarding the impact of land use-related factors on the prevalence and distribution of antimicrobial-resistant *Staphylococcaceae* in game ungulate populations is necessary for adequately determining risks related to interactions between wild animals, domestic animals, and humans in common geographic locations. This systematic approach for the analysis of the observations in specific hunting districts of Brandenburg, Germany, adds an innovative value to the research strategy of antimicrobial resistance in wild game animals, which is in accordance with current recommendations worldwide. Thus, results and information obtained in this study build a relevant foundation for future risk assessment regarding the safety of game products. Furthermore, the data generated represent an important basis for improving existing guidelines in land use practices and hunting practices. The use of existing open source data collections provided by official governmental and nongovernmental entities increases not only the impact but also the applicability and comparability of information beyond the regional level.

**KEYWORDS** antimicrobial resistance, human activities, human-animal-environment interface, fallow deer, red deer, roe deer, wild boar

**Ad Hoc Peer Reviewer** Babafela Awosile

Address correspondence to Rafael H. Mateus-Vargas, rafael.mateus-vargas@bfr.bund.de.

The authors declare no conflict of interest.

The use of antimicrobial compounds to treat and prevent infectious diseases has increased the selection pressure on microbial populations and, consequently, has contributed to the development and spread of antimicrobial-resistant bacteria (1). Infection with resistant pathogens has a direct influence on the effectiveness of subsequent antibiotic therapy, while the uptake of resistant, nonpathogenic bacteria has been recognized as a potential source of various resistance properties for human intestinal and skin microbiota due to horizontal antimicrobial resistance (AMR) gene transfer (2, 3). Regarding possible effects of human activities on natural ecosystems, the use of antimicrobial substances and the resulting presence of antimicrobial-resistant bacteria and/or antimicrobials in the environment is of increasing interest worldwide (4). From a One Health perspective, the epidemiological spread of AMR in the environment may have both direct and indirect repercussions on public health (5). In Europe, the spread of AMR in wild ungulates has important relevance, considering the ecological and phylogenetic relationships with livestock (6), as well as its importance linked to the production of food from these animals (7, 8).

Although extensive surveillance has been carried out on the occurrence of antimicrobial-resistant microorganisms in the fecal microbiota of wild game ungulates in Europe (9–25), remarkably limited information is available on the epidemiology of AMR among staphylococcal species in such populations. This is probably due to a very low occurrence of resistant *Staphylococcaceae* in European fallow deer, red deer, roe deer, and wild boar populations reported by other authors (22, 23, 26–32). On the other hand, a strict selective approach, which mainly focuses on the detection of methicillin-resistant *Staphylococcus aureus* (MRSA), may disregard the presence of AMR in other staphylococci. As shown previously in dairy herds with a history of MRSA detection, antimicrobial-resistant members of the family *Staphylococcaceae*, other than *S. aureus*, are highly prevalent, show phenotypic resistance to various antimicrobial substances, and may harbor important AMR genes (33–35). It is noteworthy that antimicrobial-resistant *Staphylococcaceae*, including *S. aureus*, have been detected in air and/or dust samples from dairy (35), pig (36, 37), and poultry farms (38) as well as in human hospitals (39). Similarly, various staphylococci, including resistant bacteria, have been detected in anthropogenic wastewater (40, 41). Since the most probable source of resistant microorganisms of fecal origin for European (12) and North American (42) wild animals is contact with farm animals, sewage, or manure, an exposure of wild game ungulates to bioaerosols, biosolids, or surface waters contaminated with resistant *Staphylococcaceae* from anthropogenic sources remains likely. In fact, Monecke et al. (28) hypothesized that the detection of MRSA in a German sample of fallow deer was related to livestock. Their hypothesis was based on the results of multilocus sequence typing (MLST) of the resistant strain as well as the sequence types of further susceptible *S. aureus* of wild game ungulates (28). Similarly, Mama et al. (43) and Sousa et al. (30) reported the isolation of a livestock-associated (LA) MRSA strain (MLST ST398) from a wild boar in the south of Spain and from one in the north of Portugal, respectively. However, Rey Pérez et al. (32) failed to find a relation between the methicillin-resistant isolates of *S. sciuri* in wild boar or red deer and either the animal source or the geographical origin in Spain.

In this matter and despite the significant value of previous studies, the generation of decisive data and information about factors influencing the transmission and persistence of AMR in wild populations remains a fundamental challenge for researchers. In particular, the degree of influence of the potential anthropogenic sources on the prevalence of antimicrobial-resistant *Staphylococcaceae* (agriculture versus residential sources) has not yet been delineated with sufficient certainty. Interestingly, Darwich et al. in Spain (20) and Formenti et al. in Italy (21), as well as Holtmann et al. in Germany (22), incorporated further data concerning the land use features of the studied regions into their analyses regarding the prevalence of antimicrobial-resistant *Escherichia coli* in fecal samples of wild boars. By using different data sources, those authors observed statistically supported correlations between the occurrence of such resistant bacteria

and the human population density of the locations in which the wild boars were sampled. The observations of the authors mentioned above show that the inclusion of land use-related data is particularly relevant for the investigation of AMR in wild ungulates in Europe. Habitat characteristics of European wild ungulates are strongly affected by the regionally varying combinations of cultural and natural landscapes, which may differently influence the degree of exposure to AMR sources as well as their distribution within game animal populations.

The aim of this study was to evaluate the occurrence of *Staphylococcaceae* showing phenotypic resistance to cefoxitin in a wild ungulate population in Brandenburg, Germany, in terms of the possible influence of human activities on their detection. Analyses were based on the use of open source data concerning the extent as well as the degree of land use, especially for settlement or animal husbandry.

## RESULTS

**Prevalence of *Staphylococcaceae* exhibiting reduced susceptibility to cefoxitin differed between species but was not influenced by season, sex, or age.** During the 3-year survey, 371 nasal swabs were obtained, of which 45 (12.1%) tested positive for *Staphylococcaceae* exhibiting a non-wild-type phenotype for cefoxitin (>4 mg/liter). The distribution of the phenotypically resistant isolates by hunting district and animal species, as well as the matrix-assisted laser desorption ionization–time of flight (MALDI-TOF) mass spectrometry identification, are summarized in Table 1. Specific data related to positive animals are summarized in Table S1 in the supplemental material. In general, the detection rates of the isolated staphylococci for fallow deer, red deer, roe deer, and wild boar were 21.8% (19/87), 22.7% (5/22), 11.5% (14/122), and 5.0% (7/140), respectively. Statistical significance was determined in this case for differences in the prevalence observed between nasal swabs of fallow deer and wild boar as well as the difference between samples of red deer and wild boar ($P = 0.0002$ and $P = 0.01$, respectively, chi-square test). However, differences in the detection rate of *Staphylococcaceae* with reduced susceptibility to cefoxitin in samples were not associated with hunting season, sex, nor age ($P > 0.05$, chi-square test).

Based on MALDI-TOF identification and irrespective of the animal source, bacteria of the genus *Mammaliicoccus* were the most frequently isolated bacteria from all game species (30/45), followed by *S. aureus* (10/45), *Staphylococcus saprophyticus* (3/45), *Staphylococcus epidermidis* (1/45), and *Staphylococcus succinus* (1/45) (Table 1). Specific information about animal carriers for every isolate is shown in Table S1.

**Regional livestock density was correlated with detection of non-wild-type *Staphylococcaceae*.** Separated by hunting districts, average prevalence rates of *Staphylococcaceae* with non-wild-type phenotype for cefoxitin were 9.0% (95% confidence interval [CI], 5.1% to 12.9%) for all game animals, 8.8% (95% CI, 0% to 23.9%) for fallow deer, 41.7% (95% CI, 0% to 90.5%) for red deer, 8.1% (95% CI, 2.2% to 14.1%) for roe deer, and 4.5% (95% CI, 0.7% to 8.3%) for wild boar. According to Spearman's correlation test, the detection of isolated *Staphylococcaceae* was only significantly associated with the hunting district of origin in red deer samples ($r_s = 0.83$, $P = 0.04$).

Based on geodetic data obtained for all selected radii, proportional land coverage in the sampled hunting districts ranged between 0.0 and 65.1% for agriculture, 30.9 and 100.0% for forest, 0.0 and 28.1% for urban, and 0.0 and 9.8% for water areas. For example, the proportional extent of land coverage within a 4.4-km radius by every land use type is shown in Fig. 1. Regarding the possible influence of land coverage on the results, only the proportional extent of water in a 2.2-km radius for roe deer ($r_s = 0.47$, $P < 0.05$) as well as the proportional extent of water in the respective municipality or county for fallow deer (both $r_s = 0.89$, $P = 0.04$) were statistically associated with the detection rate of *Staphylococcaceae* with reduced susceptibility to cefoxitin in the different hunting districts. Interestingly, a positive correlation was detected between the prevalence of phenotypically resistant *Staphylococcaceae* in samples of roe deer and either the density of cattle in the county ($r_s = 0.59$, $P = 0.01$) or the population density of the respective municipality ($r_s = 0.47$, $P < 0.05$). On the other hand, the

**TABLE 1** Number of samples per hunting district and MALDI-TOF identification of *Staphylococcaceae* showing reduced susceptibility to cefoxitin isolated from nasal swabs of game ungulates

| Hunting district | Fallow deer | | | Red deer | | | Roe deer | | | Wild boar | | |
|---|---|---|---|---|---|---|---|---|---|---|---|---|
| | Sampled animals | No. of positive individuals | Isolate species (n) | Sampled animals | No. of positive individuals | Isolate species (n) | Sampled animals | No. of positive individuals | Isolate species (n) | Sampled animals | No. of positive individuals | Isolate species (n) |
| A | 0 | | | 0 | | | 7 | 1 | *S. aureus* (1) | 1 | 0 | |
| B | 0 | | | 1 | 0 | | 6 | 0 | | 8 | 0 | |
| C | 0 | | | 0 | | | 6 | 2 | *Mammaliicoccus* sp. (2) | 4 | 0 | |
| D | 2 | 0 | | 0 | | | 5 | 1 | *Mammaliicoccus* sp. (1) | 11 | 1 | *Mammaliicoccus* sp. (1) |
| E | 1 | 0 | | 0 | | | 3 | 0 | | 1 | 0 | |
| F | 0 | | | 7 | 0 | | 7 | 0 | | 10 | 1 | *Mammaliicoccus* sp. (1) |
| G | 0 | | | 0 | | | 0 | | | 4 | 1 | *S. aureus* (1) |
| H | 0 | | | 0 | | | 6 | 0 | | 1 | 0 | |
| I | 0 | | | 4 | 1 | *Mammaliicoccus* sp. (1) | 2 | 0 | | 8 | 0 | |
| J | 58 | 14 | *Mammaliicoccus* sp. (7) *S. aureus* (5) *S. saprophyticus* (2) | 0 | | | 3 | 0 | | 16 | 0 | |
| K | 0 | | | 1 | 1 | *Mammaliicoccus* sp. (1) | 8 | 0 | | 5 | 1 | *Mammaliicoccus* sp. (1) |
| L | 0 | | | 0 | | | 3 | 0 | | 0 | | |
| M | 0 | | | 8 | 2 | *Mammaliicoccus* sp. (1) *S. epidermis* (1) | 12 | 4 | *Mammaliicoccus* sp. (3) *S. aureus* (1) | 32 | 2 | *Mammaliicoccus* sp. (2) |
| N | 0 | | | 0 | | | 10 | 1 | *S. aureus* (1) | 6 | 0 | |
| O | 0 | | | 0 | | | 22 | 4 | *Mammaliicoccus* sp. (3) *S. succinus* (1) | 12 | 0 | |
| P | 0 | | | 0 | | | 7 | 0 | | 3 | 0 | |
| Q | 22 | 5 | *Mammaliicoccus* sp. (5) | 1 | 1 | *S. aureus* (1) | 4 | 0 | | 10 | 1 | *Mammaliicoccus* sp. (1) |
| R | 4 | 0 | | 0 | | | 3 | 0 | | 1 | 0 | |
| S | 0 | | | 0 | | | 8 | 1 | *S. saprophyticus* (1) | 7 | 0 | |
| Total | 87 | 19 | | 22 | 5 | | 122 | 14 | | 140 | 7 | |

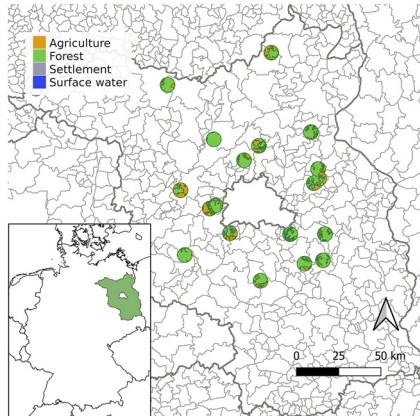

**FIG 1** Political map of the federal state of Brandenburg, Germany, indicating the distribution of hunting districts where animals were sampled. Circles around the coordinates of hunting districts show the proportional extent of land coverage within a 4.4-km radius by agriculture (orange), forest (green), settlement (gray), and surface water (blue). Boundaries between land use types are delineated in bold green. Delimitations of federal states and those of municipalities are framed with bold and pale gray lines, respectively.

prevalence in the sampled roe deer population was negatively associated to the density of pigs in the county ($r_s = -0.61$, $P = 0.007$). For wild boars, the detection rate in swab samples was positively associated with the density of poultry in the county ($r_s = 0.62$, $P = 0.006$). Additionally and in contrast to observations for roe deer, the prevalence in the wild boar population was negatively correlated with the human population in the municipalities ($r_s = -0.49$, $P = 0.04$). According to logistic regression analyses, only the significant association between the detection of *Staphylococcaceae* with a non-wild-type phenotype for cefoxitin in samples of roe deer and the density of cattle in the county (odds ratio [OR], 1.01; $P < 0.05$), as well as between the detection of these bacteria in nasal swabs of wild boar and the density of poultry in the county (OR, 1.28; $P = 0.03$) were confirmed. The resulting OR for tested predictors, respective 95% CIs, and $P$ values are shown in Table 2.

## DISCUSSION

The impact of human activities on the epidemiology of AMR in the environment is, from a holistic perspective, of great relevance for public health. In this matter, the analysis of the interactions contributing to the distribution of antimicrobial resistant bacteria between and/or within natural and human-used environments is essential. In this study, we analyzed the occurrence of *Staphylococcaceae* showing phenotypic resistance to cefoxitin in nasal swabs obtained from four important game animal species in the federal state of Brandenburg, Germany. As previously mentioned, swab sampling was performed in one or both nasal orifices by different persons. This fact is to note because the sampling of only one naris may have resulted, at least theoretically, in an

**TABLE 2** Probability of detection of *Staphylococcaceae* with reduced susceptibility to cefoxitin in nasal swab samples of wild ungulates in hunting districts located in Brandenburg, Germany, based on population or livestock densities in the respective counties

| | Fallow deer | | | Red deer | | | Roe deer | | | Wild boar | | |
|---|---|---|---|---|---|---|---|---|---|---|---|---|
| Predictor | OR | 95% CI | *P* value | OR | 95% CI | *P* value | OR[a] | 95% CI[a] | *P* value[a] | OR[a] | 95% CI[a] | *P* value[a] |
| Population in municipality[b] | 1.08 | 0.09–13.64 | 0.955 | 2.15 | 0.29–16.00 | 0.457 | 2.01 | 0.86–4.69 | 0.105 | 0.16 | 0.02–1.55 | 0.114 |
| Population in county[b] | 1.17 | 0.10–13.56 | 0.902 | 0.08 | 0.00–14.59 | 0.347 | 1.32 | 0.40–4.40 | 0.653 | 0.15 | 0.01–3.96 | 0.255 |
| Livestock density[c] | 0.71 | 0.38–1.35 | 0.295 | 0.89 | 0.75–1.06 | 0.18 | 1.01 | 0.93–1.11 | 0.76 | 1.03 | 0.93–1.15 | 0.566 |
| Cattle density[c] | 0.82 | 0.56–1.21 | 0.322 | 1.16 | 0.90–1.49 | 0.251 | **1.01** | **1.00–1.20** | **0.046** | 0.93 | 0.80–1.09 | 0.351 |
| Pig density[c] | 0.92 | 0.72–1.16 | 0.455 | 0.84 | 0.64–1.11 | 0.224 | 0.83 | 0.69–1.00 | 0.055 | 1.09 | 0.96–1.25 | 0.192 |
| Poultry density[c] | 0.95 | 0.61–1.47 | 0.802 | 0.91 | 0.68–1.20 | 0.492 | 0.92 | 0.77–1.10 | 0.354 | **1.28** | **1.03–1.59** | **0.025** |

[a]Associations with statistical significance ($P < 0.05$) are shown in bold.
[b]Population density, expressed as inhabitants per square kilometer.
[c]Livestock density, expressed as livestock units (LUs) per square kilometer.

**TABLE 3** Sizes of home ranges reported in the literature for European ungulates

| Animal species | Avg home range size (km²) | | | | | | References |
|---|---|---|---|---|---|---|---|
| | Females | | | Males | | | |
| | Q1[a] | Median | Q3[a] | Q1[a] | Median | Q3[a] | |
| Fallow deer | 2.13 | 3.21 | 4.54 | 3.07 | 4.64 | 6.86 | 47–50 |
| Red deer | 6.08 | 8.84 | 13.09 | 15.36 | 36.00 | 40.97 | 51–61 |
| Roe deer | 0.43 | 0.68 | 0.86 | 0.69 | 0.98 | 1.11 | 44–46, 62–64 |
| Wild boar | 4.56 | 6.71 | 7.50 | 6.01 | 8.39 | 10.10 | 58, 65–68 |

[a]Q1 and Q3 are the lower and upper quartiles, respectively.

underestimation of the prevalence of phenotypically resistant bacteria in the nasal microbiota of the studied wild ungulates. By concentrating the assessment on this particular federal state, we intended to avoid relevant geo-topographic factors (e.g., high variable altitudes, radical differences in resource type or resource availability, or even fundamental climate conditions), which can variably influence intra- and interspecific interactions of populations of wild game in Europe (44, 45, 46). Such factors may be expected to complicate the analyses of the influence of land use-related components on the detection rate between different wild populations of more heterogeneous landscapes. To assess the level of importance of anthropogenic factors on our results, we incorporated a systematic consideration of existing open source data collections provided by official governmental entities regarding land use extent as well as data regarding the intensity of land use for either settlement or animal husbandry. These data are annually gathered and published by the German statistical offices for the municipalities or counties. Other authors in Germany have previously used these data sets. For instance, Holtmann et al. (22) evaluated the prevalence of extended-spectrum $\beta$-lactamase-producing and AmpC-producing *Escherichia coli* in wild boars in different counties distributed across Germany. An important asset of the present study was the use of customized geodetic data provided by the European Corine Land Cover database. The sizes of reference areas for the characterization of land use features were made considering the average home range sizes reported by different authors for the four artiodactyl species in Europe (Table 3). When using data on space use of game animals, it is important to note that geographical, seasonal, and methodological factors differently influence the estimations of home ranges in these populations (44–68). Although the areas generated are an artificial approximation of the areas inhabited by the animals sampled and will factually not apply for all sampled animals, this expansion of the spatial data sources permitted uncoupling of the analyses from possible interfering influences produced by political land delineations. To the best of our knowledge, this is the first report of a systematic approach for the spatial evaluation of AMR in *Staphylococcaceae* isolated from free-living wild game ungulates.

In general, results showed that the nasal flora of sampled game animals may contain *Staphylococcaceae* with non-wild-type phenotypes for cefoxitin. As mentioned before, detection of resistant staphylococci in European wild ungulates is rare. For instance, antimicrobial-resistant *S. aureus* has been detected by other authors in a few nasal swab samples of European fallow deer (1/13 [28]), red deer (1/9 [31]), or wild boar (13/371 [43], 3/795 [27], or 1/45 [30]). Other authors failed to isolate any resistant *S. aureus* in wild ungulate species (22, 23, 26–28, 32). In the present study, it seemed appropriate to expand the analysis strategy and target *Staphylococcaceae* exhibiting reduced susceptibility to cefoxitin. The latter was used to accommodate the fact that *Staphylococcaceae*, other than *S. aureus*, may show resistance and harbor relevant genetic AMR factors, as previously shown in dairy herds (33–35). In fact, Mama et al. (43, 69) more frequently detected resistant non-*S. aureus* staphylococci than resistant *S. aureus* in swab samples of Spanish wild boars (41/371 versus 13/371). A further relevant feature of our study was the possibility to sample different game species in each hunting district. Through this strategy, we expected to obtain a wider perspective by comparing the occurrence of nonsusceptible *Staphylococcaceae* between animals

hunted within the same area. Interestingly, Rey Perez et al. (32) followed a similar approach by aiming to detect methicillin-resistant coagulase-negative staphylococci in different game species in different locations in central western Spain. Through a wider screening scheme, and despite their negative findings concerning the presence of MRSA, 2 of 42 red deer, as well as 5 of 90 wild boars, tested positive for methicillin-resistant *Staphylococcus* spp. (32). In comparison, a total of 10 *S. aureus* isolates showing reduced susceptibility to cefoxitin, and also a further 35 resistant isolates of the family *Staphylococcaceae*, were detected in the four game ungulates of the studied population in Brandenburg, Germany. These apparent coincidences may indicate similar trends. However, despite the similarities or disparities mentioned above, it is important to note that the comparisons of our results with previous studies should be interpreted with caution. This is due to the inherent methodological differences concerning the focus of examination and isolation regarding the study of antimicrobial-resistant *Staphylococcaceae* in particular wild game animals.

In the sampled hunting districts, the detection rate of *Staphylococcaceae* with a non-wild-type phenotype for cefoxitin varied between game animal species. Generally, the nasal swabs of fallow deer and red deer harbored significantly more frequently nonsusceptible bacteria than the samples of roe deer (~2-fold more prevalent) and wild boar (~4-fold more prevalent). However, it is important to note that both deer species were only sampled in a few hunting districts, and the sample size was relatively low (Table 1). In contrast, roe deer and wild boar were consistently sampled in the selected hunting districts (Table 1). Beyond the fact that these animals were hunted in specific locations and were thus probably sharing a common habitat, the studied wild artiodactyl species are, from a biological perspective, considerably distinct. This is not only based on the systematics and taxonomic classification of these animals, but also with regard to their physiology, behavior, phenology, and nutrition. From an epidemiological perspective, such differences are relevant and may influence the degree of exposure to common sources of microorganisms as well as the further distribution within animals of specific habitats. For instance, the observations of Schotte et al. (70) indicated the relevance of such biological differences on the distribution of transmissible agents within game animals in a German survey. Those authors observed that the spatial distributions of hepatitis E virus (HEV) subtypes in populations of wild ungulates were related to specific locations, which also suggested common infection sources for the sampled animals (70). On the other hand, the prevalence of HEV differed between cervids and wild boar which, according to the authors, indicated different exposure rates and/or different transmission dynamics of this virus (70). Despite the obvious (epidemiological) differences between viruses and bacteria, similar observations were previously made by comparing the prevalence of MRSA in free-living populations of European rabbit, red deer, mouflon, and wild boar in northern Spain (31). Thus, the observations made in our study may support the relevance of such biological characteristics on the detection rate of *Staphylococcaceae* with reduced susceptibility to cefoxitin in different wild game species in the same hunting districts. The localized sampling of both fallow and red deer in the sampled area and the sample size, as well as the fundamental biological differences between fallow deer, red deer, and wild boar, complicate further statements regarding the statistically significant differences in the detection rates of *Staphylococcaceae* exhibiting a non-wild-type phenotype for cefoxitin. In the future, studies should consider these aspects to explore regional associations and draw implications about the possible consequences for such populations.

Regarding the degree of influence of anthropogenic activities on our results, statistical analyses showed that the intensity of land use, particularly for animal husbandry, may influence the detection rate of *Staphylococcaceae* with a non-wild-type phenotype for cefoxitin in the game ungulates inhabiting a specific region. These observations contrast with the results reported for $\beta$-lactam-resistant *Escherichia coli*, for which the human population density was assumed to importantly affect the prevalence in European wild boars (20–22). Although the results were not completely foreseeable, we expected to observe a notable influence of livestock on detection rates. Without

consideration of spatial data, Porrero et al. (27) hypothesized that livestock may be a source of MRSA for free-living wild animals. In fact, more recent reports regarding the presence of resistant staphylococci were made in game ungulates of a Spanish region that is characterized by agricultural activities, especially animal holdings of pigs and ruminants (30). In another Spanish study, MRSA was only detected in a wild boar that was hunted in a region with several Iberian pig farms (43). Remarkably, a statistically significant correlation was consistently determined by our analyses between the detection rate of nonsusceptible *Staphylococcaceae* in roe deer or wild boar and the density of two particular domestic animal populations, cattle and poultry. Thus, it can be assumed that the inherent divergences between roe deer and wild boar, which go beyond the physiological aspects (e.g., extent of habitat use by single animals) (Table 3), may considerably influence the colonization rate of the particular artiodactyl species to common sources of antimicrobial-resistant *Staphylococcaceae.* Furthermore, it is possible to hypothesize that the differences in bacterial concentrations in bioaerosols between specific animal facilities (35–38) may pose different threats to the surrounding ecosystems regarding the emission of resistant staphylococcal bacteria. Although the regional proportion of land used by humans, including agriculture and settlement, did not seem to explain our findings, it was interesting to detect single positive correlations between the extent of surface waters and the detection rate of non-wild-type bacteria in nasal swabs of roe deer and fallow deer. Although this observation was not confirmed by the logistic regressions, it was especially interesting to find this statistically significant correlation to the proportional extent of surface water in a relatively small area (within a 2.2-km radius, i.e., an area of 15 km$^2$) for roe deer, which is also the free-living ungulate species with the smallest average home range size in Europe ($<$2 km$^2$) (Table 3). Since the presence of MRSA in drinking water of livestock (71) and subsequent survival capacity in wastewater (40, 41) have been confirmed in previous studies, it is plausible that surface water may represent a vector and source of resistant *Staphylococcaceae* for at least some ungulate species. Combined with a high livestock density, animals in regions with larger water bodies may be at higher risk for exposure to such resistant bacteria. In the future, risk-oriented surveillance should permit further assessment of these assumptions.

In conclusion, it was determined that the nasal flora of the sampled species of wild ungulates of Brandenburg, Germany, harbored *Staphylococcaceae* exhibiting reduced susceptibility to cefoxitin. This observation is of particular interest, since it is currently hypothesized that members of *Staphylococcaceae* may serve as a reservoir of AMR genes for more-pathogenic *S. aureus* strains (72). Additionally, through the presented approach, a spatial evaluation of the detection rate of nonsusceptible *Staphylococcaceae* in the free-living game population permitted relevant assessments from a local perspective. Considering the limitations of the sampling strategy, the presented results reflect the regional situation of the hunted game species (73). According to our observations, a high density of livestock and the presence of large surface water bodies may increase the local chances of detecting antimicrobial-resistant *Staphylococcaceae* in wild game animal populations. Following the premise of methodic standardization, the use of geodetic information may represent an important method to complement efforts to understand the processes influencing the exposure and colonization of these animals during their lifetime until harvest. Considering the divergences in the detection rates between the wild artiodactyl species and based on the cited literature, generalizations of observations regarding the prevalence of transmissible agents in wild game animals (even within cervids) should be performed with caution. In this matter, although these animals may have shared a common habitat, divergences between animal species, including home range sizes, may have affected the chances and/or degree of exposure to anthropogenic sources for specific populations and therefore require a differentiated consideration. Despite the redundant analyses of artificial areas generated with different radii, the gradient of custom ranges seemed adequate to avoid either overestimation (especially for roe deer data) or underestimation (especially for red deer data) of land use-related factors possibly influencing the observations.

Based on the results, it is generally recommendable that more than one animal species showing sufficient biologic differences should be sampled to detect different spread patterns and pathways in common habitats. Despite the relevance of wild boars as a bioindicator for surveillance of the prevalence and spread of AMR in different ecosystems (74), it seems adequate to consider roe deer as a widely usable biologic coindicator, especially in environments dominated by various human activities in Europe. The use of further species such as fallow deer may regionally be relevant due to its localized dissemination and comparable sizes of home ranges (Table 3). In the case of red deer, both the greater interspecific variations of home range sizes (Table 3) as well as the different spatial and temporal migration patterns of specific European populations (55, 56) may pose considerable challenges during the interpretation of such epidemiologic data at the regional, national, and continental levels. Overall, data on the prevalence and distribution of AMR in game populations accompanied by the consideration of land use-related factors may have an important impact on the adequate determinations of risk factors linked to the interactions between wild animals, domestic animals, and humans. Moreover, phylogenetic examination of susceptible and non-susceptible *Staphylococcaceae* will complement the presented approach regarding the transmission pathways and distributions of resistant staphylococcal bacteria in wild game ungulates. Due to the wide accessibility of data regarding land use extent and intensity, the adaptability, applicability, and comparability of our observations and the hypothesis presented above should be addressed through risk-oriented interdisciplinary approaches, at least at the national and European continent levels.

## MATERIALS AND METHODS

**Study area and sampling strategy.** Sampling was conducted in 19 different hunting districts of the German federal state of Brandenburg (Fig. 1). This federal state is located in the east of Germany, at the border to Poland, and has an area of approximately 29,654 km². From a topographical perspective, the landscape of Brandenburg is characterized by large flat land areas and localized low-altitude hills, with the Heidehöhe of Heideberg the highest elevation (201.4 m above sea level) (https://geobasis-bb.de/sixcms/media.php/9/Land -Brandenburg-in-Zahlen-und-Karten.pdf [last accessed 31 March 2022]). Without considering the constituent state of Berlin, approximately 2,037 km² of the total area of this federal state consist of human settlement (6.9%), 14,426 km² consist of agricultural land (48.6%), 10,320 km² consist of forest stand (34.8%), and 998 km² consist of surface water (3.4%) (based on data for 2020, available at https://www.statistischebibliothek.de [last accessed 31 March 2022]). Sampling activities were conducted within a framework agreement involving the German Federal Institute for Risk Assessment (BfR) and the German Institute for Federal Real Estate (BImA), as previously described in detail by Maaz et al. (73). For this study, nasal swabs were taken from healthy free-living fallow deer (*Dama dama*), red deer (*Cervus elaphus*), roe deer (*Capreolus capreolus*), and wild boar (*Sus scrofa*) hunted at drive hunts in late autumn and winter during three consecutive hunting seasons from 2019-2020 to 2021-2022. Wild ungulates were legally hunted within the population management program of the German Federal Forest Service. Therefore, official approval was neither required for the manipulation of carcasses nor for their sampling, and legal requirements regarding ethical standards were fulfilled. Carcasses to be sampled were chosen randomly, and the number of animals sampled depended on the bag of the day (73). After each hunt, a single swab sample was taken from one or both nasal orifices of game either by hunters or by scientific staff. Nasal swabs were cooled and transported to the facilities of the BfR, where they were stored at 4°C until microbiological analysis.

**Microbiological analyses.** Isolation of *Staphylococcaceae* exhibiting reduced susceptibility to cefoxitin from nasal swabs was performed using a double-selective enrichment method, which was originally developed for MRSA detection and with which excellent results have been achieved for other *Staphylococcaceae* (35). Swab samples were incubated in 10 mL Mueller-Hinton (MH) broth supplemented with 6% NaCl for 24 ± 2 h. An overnight culture (1 mL) was transferred to 9 mL of tryptic soy broth supplemented with 3.5 mg/liter cefoxitin and 50 mg/liter aztreonam and incubated for 24 ± 2 h at 37°C. The enrichment broth (50 µL) was streaked on mannitol salt agar (MSA) containing 4 mg/liter cefoxitin and incubated for 24 ± 2 h at 37°C. Cefoxitin concentration of selective plates was chosen according to the epidemiological cutoff value (ECOFF value) reported by the European Committee on Antimicrobial Susceptibility for *Staphylococcus aureus* (https://www.eucast.org/mic_distributions_and _ecoffs/ [last accessed 4 July 2022]). All colonies from MSA-cefoxitin plates were transferred on sheep blood agar plates (Oxoid GmbH, Wesel, Germany) and incubated for 24 ± 2 h. Colonies from sheep blood agar plates were identified by a MALDI-TOF mass spectrometer according to the manufacturer's instructions (Bruker Scientific LLC, Billerica, MA). MALDI-TOF MS is frequently used for microbial identification (75). For MALDI-TOF MS, microbial samples were crystallized within a matrix and ionized by a laser beam. Protonated ions were analyzed according to their mass-to-charge ratio and compared to known databases for identification of microorganisms. Colonies were directly transferred on the MALDI-TOF target and covered with 1 µL of α-cyano-4-hydroxycinnamic acid (Bruker Scientific LLC). The reference database for species identification was provided by Bruker Scientific LLC (MBT-BDAL-8468). If

phenotypically different colonies were observed on sheep blood agar plates, they were separately spotted on the MALDI-TOF target.

**Characterization of hunting districts and statistical analyses.** At first, the coordinates corresponding to the middle point of each hunting district were determined. Estimated coordinates built the reference points for further analyses. Hunting districts were characterized by land use types and their extent as well as the intensity of use according to the following strategy. Initially, geographic coordinates representing the estimated geometric center of each hunting district were used to assign them to a specific municipality and county. Characterization was then performed by using data sets describing the relative extent of land use for settlement and agriculture as well as the relative extent of areas covered by forest or surface water at a municipal or county level (as a percentage). Additionally, data were included regarding population density (people per square kilometer) in the respective municipalities and counties as well as the density of livestock animals in the counties. Livestock density was calculated using the information about the extent of land used for animal husbandry and the respective livestock units (LUs) of cattle, pigs, and poultry reported for the county (LU per square kilometer). Data on land features, population density, and LUs were those gathered and published by the Berlin-Brandenburg Statistics Office on its official website (based on data for 2020; available at https://www.statistischebibliothek.de [last accessed 31 March 2022]). In order to address possible weaknesses of this approach (e.g., a type I error regarding land use extent in large municipalities or counties), the characterization strategy was complemented by using further data on land use-related components obtained from the Corine Land Cover (CLC) database in its 2018 version (data for 2018 are available at https://land.copernicus.eu/pan-european/corine-land-cover/clc2018 [last accessed 14 February 2022]). Using the QGIS software (version 3.12.0, București; QGIS.org), the number of original land use categories contained in the CLC data set was dissolved by aggregating classes, ultimately distinguishing urban and agricultural areas, as well as the areas covered by forest and surface water. The aggregated land cover data were then intersected with areas generated from each coordinate within a 0.8-, 1.3-, 1.6-, 1.9-, 2.2-, 3.1-, 3.8-, or 4.4-km radius to calculate the relative extent of each land use type for each hunting district. The different radii were chosen according to a literature review regarding the average size of home ranges reported in European countries for the four sampled game animal species, whereby a circular area was assumed for the sake of simplicity (Table 3).

Statistical calculations were performed using SAS software version 9.4 for windows (SAS Institute Inc., Cary, NC, USA). The results of the statistical tests with a $P$ value of $<0.05$ were considered significant. Initially, the MEANS procedure was used to calculate descriptive statistics regarding the average prevalence by hunting district, for all animals or separated by animal species, and the respective 95% confidence intervals (95% CI). The variables species, age, sex, and hunting season were analyzed using a chi-square test (PROC FREQ; CHISQ) to determine whether differences in detection rates of targeted *Staphylococcaceae* in swab samples were statistically significant between groups for each variable. Using the characterization made with data of the municipalities, counties, or CLC, Spearman's rank correlation coefficient (PROC CORR) was used to statistically evaluate the degree of correlation of the proportional extent of land covered by settlements, agricultural land, forest, or surface water (as percentages) and the detection rate of non-wild-type *Staphylococcaceae* in swab samples. These calculations were conducted for all ungulates together as well as for each animal species separately. Furthermore, Spearman's correlation test as well as logistic regression analyses (PROC LOGISTIC) were performed to test the effects of the population density or the density of food-producing animals in the respective regions on the detection rates in nasal swabs of each wild animal species separately.

## SUPPLEMENTAL MATERIAL

Supplemental material is available online only.
**SUPPLEMENTAL FILE 1**, PDF file, 0.4 MB.

## ACKNOWLEDGMENTS

We sincerely thank the employees of the Federal Forest Division of the BImA and each individual hunter who participated in sampling activities. Additionally, we are grateful to the colleagues at BfR, especially S. Sutrave, K. Stollberg, and C. Kästner, for their collaboration during the organization and performance of sampling activities. We furthermore thank M. Kirchner for the digitalization of data and S. F. Marino for sample collection and the editing of the manuscript.

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
