## [Reviewer comments · Microbiology Spectrum]

Microbiology Spectrum

Evaluation of the occurrence of Staphylococcaceae showing reduced susceptibility to ceftiofur in wild ungulates in Brandenburg, Germany, based on land use-related factors

Rafael Mateus Vargas, Tobias Lienen, Denny Maaz, Martin Richter, Sven Maurischat, and Julia Steinhoff-Wagner

Corresponding Author(s): Rafael Mateus Vargas, German Federal Institute for Risk Assessment (BfR)

Review Timeline:

Submission Date:	July 5, 2022
Editorial Decision:	August 4, 2022
Revision Received:	August 24, 2022
Editorial Decision:	September 9, 2022
Revision Received:	September 10, 2022
Accepted:	September 14, 2022

Editor: Adelumola Oladeinde

Reviewer(s): Disclosure of reviewer identity is with reference to reviewer comments included in decision letter(s). The following individuals involved in review of your submission have agreed to reveal their identity: Babafela Awosile (Reviewer #1)

Transaction Report:

DOI: <https://doi.org/10.1128/spectrum.02560-22>

August 4, 2022

Dr. Rafael Hernan Mateus Vargas
German Federal Institute for Risk Assessment (BfR)
Max-Dohrn-Straße 8-10
Berlin D-10589
Germany

Re: Spectrum02560-22 (Evaluation of the occurrence of Staphylococcaceae showing reduced susceptibility to cefoxitin in wild ungulates in Brandenburg, Germany, based on land use-related factors)

Dear Dr. Rafael Hernan Mateus Vargas:

Thank you for submitting your manuscript to Microbiology Spectrum. As you will see your paper is very close to acceptance. Please modify the manuscript along the lines I have recommended. As these revisions are quite minor, I expect that you should be able to turn in the revised paper in less than 30 days, if not sooner. If your manuscript was reviewed, you will find the reviewers' comments below.

When submitting the revised version of your paper, please provide (1) point-by-point responses to the issues I raised in your cover letter, and (2) a PDF file that indicates the changes from the original submission (by highlighting or underlining the changes) as file type "Marked Up Manuscript - For Review Only". Please use this link to submit your revised manuscript. Detailed instructions on submitting your revised paper are below.

Link Not Available

Sincerely,

Adelumola Oladeinde

Reviewer comments:

Reviewer #1 (Comments for the Author):

I don't see any reason why the authors did correlation and regression together. They could have just applied either one to drive home the point.

Reviewer #2 (Comments for the Author):

Summary: The study evaluates occurrence of Staphylococcaceae with cefoxitin resistance in four (4) wild ungulate species and potential influences of human land use activities. Using correlation and binomial statistics for support, the authors conclude that there may be a spatial association between occurrence and proportion of surface waters, as well as an association between occurrence and cattle and/or poultry density. Land use associations may influence risk factors linked to interactions between wild and domestic animals, and humans, thus, interdisciplinary risk-oriented approaches should be utilized to characterize occurrence of antimicrobial resistance in wild game populations.

Line 141-Figure 1. Please include a legend on Figure 1 detailing the colors described in the figure caption.

Lines 208-210. Please provide more detail describing the intent and assumptions of the Chi-squared assessment. e.g., Chi-squared tests whether there are statistical differences between expected and observed frequencies of occurrence in each group.

Lines 226-229. The results section reports there were significant differences in the prevalence of nasal swabs in fallow deer and wild boar, and red deer and wild boar. What are the implications of this association? There is mention of using boar and roe deer as a bioindicator in the discussion (lines 422 to 246), but the results presented are not discussed for this study?

Results Section: Land use Characterization: A summary of the relative proportions of each hunting district land use type (described in lines 192-201) used should be reported either in a table or in the text. e.g., District XX-XX to XX % Agriculture, XX to XX % Forest.

Minor Editor comments:

Like reviewer 1 mentioned, is there a reason why the authors reported the results of both regression and correlation analysis?

Line 66 - 108: Too long of a paragraph. Please divide into two paragraphs.

Ensure that results presented are discussed. For instance, the authors reported significant differences in detection rates between Deer and Boar. I would recommend that the authors discuss the potential implications of this result.

Preparing Revision Guidelines

- point-by-point responses to the issues I raised in your cover letter
- Upload a compare copy of the manuscript (without figures) as a "Marked-Up Manuscript" file.
- Each figure must be uploaded as a separate file, and any multipanel figures must be assembled into one file.
- Manuscript: A .DOC version of the revised manuscript
- Figures: Editable, high-resolution, individual figure files are required at revision, TIFF or EPS files are preferred

Please return the manuscript within 60 days; if you cannot complete the modification within this time period, please contact me. If you do not wish to modify the manuscript and prefer to submit it to another journal, please notify me of your decision immediately so that the manuscript may be formally withdrawn from consideration by Microbiology Spectrum.

Review of Evaluation of the occurrence of *Staphylococcaceae* showing reduced susceptibility to ceftiofur in wild ungulates in Brandenburg, Germany, based on land use-related factors.

Summary: The study evaluates occurrence of *Staphylococcaceae* with ceftiofur resistance in four (4) wild ungulate species and potential influences of human land use activities. Using correlation and binomial statistics for support, the authors conclude that there may be a spatial association between occurrence and proportion of surface waters, as well as an association between occurrence and cattle and/or poultry density. Land use associations may influence risk factors linked to interactions between wild and domestic animals, and humans, thus, interdisciplinary risk-oriented approaches should be utilized to characterize occurrence of antimicrobial resistance in wild game populations.

Line 141—Figure 1. Please include a legend on Figure 1 detailing the colors described in the figure caption.

Lines 208-210. Please provide more detail describing the intent and assumptions of the Chi-squared assessment. e.g., Chi-squared tests whether there are statistical differences between expected and observed frequencies of occurrence in each group.

Lines 226-229. The results section reports there were significant differences in the prevalence of nasal swabs in fallow deer and wild boar, and red deer and wild boar. What are the implications of this association? There is mention of using boar and roe deer as a bioindicator in the discussion (lines 422 to 446), but the results presented are not discussed for this study?

Results Section: Land use Characterization: A summary of the relative proportions of each hunting district land use type (described in lines 192-201) used should be reported either in a table or in the text. e.g., District XX—XX to XX % Agriculture, XX to XX % Forest.

Identify Risks –
Protect Health

German Federal Institute for Risk Assessment

German Federal Institute for Risk Assessment • PO Box 12 69 42 • 10609 Berlin

Dr. Adelumola Oladeinde
Editor, Microbiology Spectrum
American Society for Microbiology Journals

German Federal Institute for Risk Assessment
PO Box 12 69 42
10609 Berlin, GERMANY
Phone +49 30 18412-0
Fax +49 30 18412-99099
bfr@bfr.bund.de
www.bfr.bund.de/en

Reference number and date of original message	Reference number (please include in reply)	Phone extension/ fax number	Date	Unit/contact
		-28904	15.08.2022	8SZ / Dr. Mateus Vargas

Response to Reviewers 1, 2, and to Editor comments

Dear Dr. Adelumola Oladeinde,

on behalf of all co-authors, I thank both reviewers for their constructive remarks on the article named „*Evaluation of the occurrence of Staphylococcaceae showing reduced susceptibility to ceftiofur in wild ungulates in Brandenburg, Germany, based on land use-related factors*“. We have implemented in the revised manuscript all suggestions for improvement whenever appropriate. Responses are placed below every comment of the review report and are formatted in italics for better identification in the text.

We hope that the improvements will meet the reviewers’ as well as the editor’s expectations.
Sincerely,

Rafael H. Mateus-Vargas

Reviewer 1

I don’t see any reason why the authors did correlation and regression together. They could have just applied either one to drive home the point.

Thank you for your comment. We decided to use both the correlation and the binomial regression to perform the statistical analyses. Due to the differences in the distribution of detection rates when comparing hunting districts, it seemed appropriate in this case to offset the fundamental limitations of both statistical tests in terms of sensitivity and specificity. Thus, we report the observations made with both statistical tests. This approach permitted us to critically discuss the results obtained, and gives the readers an overview of all information for their own further interpretation of results.

Reviewer 2

Summary: The study evaluates occurrence of *Staphylococcaceae* with cefoxitin resistance in four (4) wild ungulate species and potential influences of human land use activities. Using correlation and binomial statistics for support, the authors conclude that there may be a spatial association between occurrence and proportion of surface waters, as well as an association between occurrence and cattle and/or poultry density. Land use associations may influence risk factors linked to interactions between wild and domestic animals, and humans, thus, interdisciplinary risk-oriented approaches should be utilized to characterize occurrence of antimicrobial resistance in wild game populations.

Line 141-Figure 1. Please include a legend on Figure 1 detailing the colors described in the figure caption.

Thank you for your remark. We have included a legend on Figure 1 to detail the colors used in the figure caption.

Lines 208-210. Please provide more detail describing the intent and assumptions of the Chi-squared assessment. e.g., Chi-squared tests whether there are statistical differences between expected and observed frequencies of occurrence in each group.

Thank you for your recommendation. We have rephrased the sentence regarding the use of chi-square as follows:

Now lines 206-209: "Considering the variables: species, age, sex, and hunting season, chi-square (X^2) test was performed (PROC FREQ; CHISQ) to determine whether differences in detection rate of targeted Staphylococcaceae in swab samples were statistically significant between groups of each variable."

Lines 226-229. The results section reports there were significant differences in the prevalence of nasal swabs in fallow deer and wild boar, and red deer and wild boar. What are the implications of this association? There is mention of using boar and roe deer as a bioindicator in the discussion (lines 422 to 426), but the results presented are not discussed for this study?

Thank you for your comment. In our study, it was not possible to make further assumptions about the statistically significant differences between the deer species and wild boar due to the limitations related to the localized dissemination and sampling of fallow and red deer, sample size of both cervids, as well as the fundamental biological differences of these animal species. We rephrased the following sentences to emphasize this fact of our study:

Now Lines 356-365: "Thus, the observations made in our study may support the relevance of such biological characteristics on the detection rate of Staphylococcaceae with reduced susceptibility to cefoxitin in different wild game species in the same hunting districts. The localized sampling of both fallow and red deer in the sampled area, the sample size, as well as the fundamental biological differences between fallow deer, red deer, and wild boar complicate further statements regarding the statistically significant differences in the detection rate of Staphylococcaceae exhibiting non-wildtype phenotype for cefoxitin. In the future, studies should consider these aspects to explore regional associations and draw implications about the possible consequences for those populations."

Regarding the use of bioindicators, we recommend using roe deer and wild boar principally because of their wider distribution on the regional, national and European level. To bring this aspect into perspective, we have modified the sentences referring to the bioindicator as follows:

Now lines 425-436: Based on the results, it is generally recommendable that more than one animal species showing sufficient biologic differences should be sampled to detect different spread patterns/pathways in common habitats. Despite the relevance of wild boars as a bioindicator for surveillance of the prevalence and spread of AMR in different ecosystems (75), it seems adequate to consider roe deer as a widely usable biologic co-indicator, especially in environments dominated by various human activities in Europe. The use of further species such as fallow deer may regionally be relevant due to its localized dissemination and comparable size of home ranges (Table 1). In the case of red deer, both the greater interspecific variations of home range sizes (Table 1), as well as the different spatial and temporal migration patterns of specific European populations (55,56), may pose considerable challenges during the interpretation of such epidemiologic data at the regional, national, and continental level.

Results Section: Land use Characterization: A summary of the relative proportions of each hunting district land use type (described in lines 192-201) used should be reported either in a table or in the text. e.g., District XX-XX to XX % Agriculture, XX to XX % Forest. in the manuscript entitled

Thank you for your recommendation. Following the premise of a multidimensional approach, characterization of hunting districts resulted in extensive datasets, which cannot be summarized. Furthermore, the anonymization of hunting districts is at the core of the collaborative work with the federal forest division and direct publication of whole datasets may threaten this cooperation. Thus, we decided not to present specific data related to the hunting districts in the manuscript. However, as stated during the submission process, the data of this study can be made available on a duly motivated request from the corresponding author.

Nevertheless, considering the importance of your recommendation, we have decided to add following sentences to give an overview of the land use characteristics of hunting districts sampled in this study:

Lines 244-248: Based on geodetic data obtained for all selected radii, proportional land coverage ranged in the sampled hunting districts between 0.0 and 65.1% for agriculture, 30.9 and 100.0% for forest, 0.0 and 28.1% for urban, and between 0.0 and 9.8% for water areas. Exemplary, the proportional extent of land coverage within a 4.4 km radius by every land use type is shown in Figure 1.

Editor

Like reviewer 1 mentioned, is there a reason why the authors reported the results of both regression and correlation analysis?

Thank you for your comment. As stated before, we decided to use both the correlation and the binomial regression to perform the statistical analyses. Due to the differences in the distribution of detection rates when comparing hunting districts, it seemed appropriate in this case to offset the fundamental limitations of both statistical tests in terms of sensitivity and specificity. Thus, we report the observations made with both statistical tests. This approach permitted us to critically discuss the results obtained, and gives the readers an overview of all information for their own further interpretation of results.

Line 66 - 108: Too long of a paragraph. Please divide into two paragraphs.

Thank you for this remark. We have divided the mentioned paragraph of the introduction in two parts.

Ensure that results presented are discussed. For instance, the authors reported significant differences in detection rates between Deer and Boar. I would recommend that the authors discuss the potential implications of this result.

Thank you for your suggestion. As we explained before, we are cautious with the assumptions about possible implications of the detected significant differences between deer species and wild boar. The latter considering the differences related to sampled areas, sample size, and the fundamental biological differences between the species. We rephrased the following sentences to emphasize this fact of our study:

Now Lines 356-365: "Thus, the observations made in our study may support the relevance of such biological characteristics on the detection rate of Staphylococcaceae with reduced susceptibility to cefoxitin in different wild game species in the same hunting districts. The localized sampling of both fallow and red deer in the sampled area, the sample size, as well as the fundamental biological differences between fallow deer, red deer, and wild boar complicate further statements regarding the statistically significant differences in the detection rate of Staphylococcaceae exhibiting non-wildtype phenotype for cefoxitin. In the future, studies should consider these aspects to explore regional associations and draw implications about the possible consequences for those populations."

Based on our observations and comments regarding the sampling and biological aspects, we have modified the sentences referring to the use of these animal species as bioindicators. The sentence is now described as follows:

Now lines 425-436: Based on the results, it is generally recommendable that more than one animal species showing sufficient biologic differences should be sampled to detect different spread patterns/pathways in common habitats. Despite the relevance of wild boars as a bioindicator for surveillance of the prevalence and spread of AMR in different ecosystems (75), it seems adequate to consider roe deer as a widely usable biologic co-indicator, especially in environments dominated by various human activities in Europe. The use of further species such as fallow deer may regionally be relevant due to its localized dissemination and comparable size of home ranges (Table 1). In the case of red deer, both the greater interspecific

variations of home range sizes (Table 1), as well as the different spatial and temporal migration patterns of specific European populations (55,56), may pose considerable challenges during the interpretation of such epidemiologic data at the regional, national, and continental level.

September 9, 2022

Dr. Rafael Hernan Mateus Vargas
German Federal Institute for Risk Assessment (BfR)
Max-Dohrn-Straße 8-10
Berlin D-10589
Germany

Re: Spectrum02560-22R1 (Evaluation of the occurrence of Staphylococcaceae showing reduced susceptibility to ceftiofur in wild ungulates in Brandenburg, Germany, based on land use-related factors)

Dear Dr. Rafael Hernan Mateus Vargas:

Thank you for submitting your manuscript to Microbiology Spectrum. As you will see your paper is very close to acceptance. Please modify the manuscript along the lines I have recommended. As these revisions are quite minor, I expect that you should be able to turn in the revised paper in less than 30 days, if not sooner. If your manuscript was reviewed, you will find the reviewers' comments below.

When submitting the revised version of your paper, please provide (1) point-by-point responses to the issues raised by the reviewers as file type "Response to Reviewers," not in your cover letter, and (2) a PDF file that indicates the changes from the original submission (by highlighting or underlining the changes) as file type "Marked Up Manuscript - For Review Only". Please use this link to submit your revised manuscript. Detailed instructions on submitting your revised paper are below.

Link Not Available

Sincerely,

Adelumola Oladeinde

Editor comments:

Please divide the results into 2-3 sub-sections with meaningful headers. This will make it flow better and easier to read.

For example -

(1) Prevalence of ceftiofur resistant Staphylococcaceae was not influenced by season, sex or age;

(2) Proximity to livestock was correlated with the detection of ceftiofur resistant Staphylococcaceae

Also, can lines 408 - 451 be split into two paragraphs? It is a long paragraph as it stands.

Preparing Revision Guidelines

- Point-by-point responses to the issues raised by the reviewers in a file named "Response to Reviewers," NOT IN YOUR COVER LETTER.
- Upload a compare copy of the manuscript (without figures) as a "Marked-Up Manuscript" file.
- Each figure must be uploaded as a separate file, and any multipanel figures must be assembled into one file.

- Manuscript: A .DOC version of the revised manuscript
- Figures: Editable, high-resolution, individual figure files are required at revision, TIFF or EPS files are preferred

Please return the manuscript within 60 days; if you cannot complete the modification within this time period, please contact me. If you do not wish to modify the manuscript and prefer to submit it to another journal, please notify me of your decision immediately so that the manuscript may be formally withdrawn from consideration by Microbiology Spectrum.

Editor

Please divide the results into 2 -3 sub-sections with meaningful headers. This will make it flow better and easier to read.

For example -

(1) Prevalence of cefoxitin resistant Staphylococcaceae was not influenced by season, sex or age;

(2) Proximity to livestock was correlated with the detection of cefoxitin resistant Staphylococcaceae

Thank you for your suggestion. We divided the results section into two sub-sections. Additionally, we repositioned the paragraph concerning the average prevalence separated by hunting districts (old version lines 230-235) in the second sub-section to keep an understandable order of ideas (now lines 241-246). The headers were phrased as follows:

(1) Prevalence of Staphylococcaceae exhibiting reduced susceptibility to cefoxitin differed between species but was not influenced by season, sex or age

(2) Regional livestock density was correlated with the detection of non-wild type Staphylococcaceae

Also, can lines 408 - 451 be split into two paragraphs? It is a long paragraph as it stands.

Thank you for this remark. The paragraph was indeed too long. We have divided the mentioned paragraph of the discussion in two parts.

September 14, 2022

Dr. Rafael Hernan Mateus Vargas
German Federal Institute for Risk Assessment (BfR)
Max-Dohrn-Straße 8-10
Berlin D-10589
Germany

Re: Spectrum02560-22R2 (Evaluation of the occurrence of Staphylococcaceae showing reduced susceptibility to cefoxitin in wild ungulates in Brandenburg, Germany, based on land use-related factors)

Dear Dr. Rafael Hernan Mateus Vargas:

Your manuscript has been accepted, and I am forwarding it to the ASM Journals Department for publication. You will be notified when your proofs are ready to be viewed.

Sincerely,

Adelumola Oladeinde
Editor, Microbiology Spectrum

Journals Department
Supplemental File 1: Accept